# Randomized Clinical Trial Evaluating *AcceptME*—A Digital Gamified Acceptance and Commitment Early Intervention Program for Individuals at High Risk for Eating Disorders

**DOI:** 10.3390/jcm11071775

**Published:** 2022-03-23

**Authors:** Maria Karekla, Patrisia Nikolaou, Rhonda M. Merwin

**Affiliations:** 1Department of Psychology, University of Cyprus, Nicosia 1678, Cyprus; nikolaoupatrisia@hotmail.com; 2Department of Psychiatry and Behavioral Sciences, Duke University Medical Center, Durham, NC 27710, USA; rhonda.merwin@duke.edu

**Keywords:** early intervention, high risk for eating disorders, Acceptance and Commitment Therapy, gamification, digital intervention, vicarious learning

## Abstract

Eating disorders (ED) constitute a serious public health issue affecting predominantly women and appearing typically in adolescence or early adulthood. EDs are extremely difficult to treat, as these disorders are ego-syntonic, and many patients do not seek treatment. It is vital to focus on the development of successful early-intervention programs for individuals presenting at risk and are on a trajectory towards developing EDs. This study is a randomized controlled trial evaluating an innovative digital gamified Acceptance and Commitment early-intervention program (*AcceptME*) for young females showing signs and symptoms of an ED and at high risk for an ED. Participants (*n* = 92; *Mage* = 15.30 years, *SD* = 2.15) received either *AcceptME* (*n* = 62) or a waitlist control (*n* = 30). Analyses indicated that the *AcceptME* program effectively reduced weight and shape concerns with large effects when compared to waitlist controls. Most participants scored below the at-risk cut-off (WCS score < 52) in the *AcceptME* at end-of-intervention (57.1%) compared to controls (7.1%), with odds of falling into the at-risk group being 14.5 times higher for participants in the control group. At follow-up, 72% of completers reported scores below the at-risk cut-off in the *AcceptME* group. The intervention also resulted in a decrease in ED symptomatology and increased body image flexibility. Overall, results suggest that the *AcceptME* program holds promise for early-intervention of young women at risk for developing an ED.

## 1. Introduction

Prevention and early intervention programs for eating disorders (ED) to date have focused on either reducing the pursuit of the thin ideal [1] or on disputing and replacing unrealistic thoughts with regard to food, body, and weight [2]. These programs are based on the most influential cognitive behavioural theory of ED and conceptualized as cognitive problems [3]. According to this approach, it is the individual’s set of dysfunctional beliefs regarding the significance of weight and shape and an ED filter that biases the perspective of the world that lead to pathology [3]. The Cognitive Behavior Therapy (CBT) model of therapy focuses directly on disputing and replacing unrealistic thoughts with regard to food, body, and weight [2]. An alternative approach is to consider the function of ED symptoms and specifically the way in which they allow for avoidance and escape from unwanted internal experiences (e.g., negative affect generally and specifically related to one’s body) through symptom engagement and direct manipulation of the body [4,5,6]. Instead of attempting to alter cognitions, this approach would suggest teaching individuals to accept difficult thoughts and feelings and engage in effective action (e.g., healthy eating) in the presence of emotions or other unwanted internal experiences [5]. 

Acceptance and Commitment Therapy (ACT) [7,8] is a third-wave CBT that targets ineffective internal control strategies and inflexibility found in a range of psychological problems. Rather than attempting to change internal experiences, such as thoughts, sensations, and feelings [8], ACT aims to change how individuals relate or respond to their thoughts/feelings. The aim is greater psychological flexibility or to have unwanted internal experiences without unnecessary attempts to avoid or escape and without these experiences exerting undue influence over behaviour. ACT facilitates psychological flexibility via six core therapeutic processes: (1) experiential acceptance, (2) cognitive defusion, (3) present moment awareness, (4) self as context, (5) values clarification and living based on chosen values, and (6) committed purposeful action [9]. 

Accumulating evidence suggests that ACT holds potential for the treatment and prevention of EDs [4,10,11,12]. ACT has been shown to have efficacy for reducing ED symptoms relative to a waitlist control and treatment as usual (TAU) [13,14] and to have greater effects compared to cognitive therapy in one trial of ED symptoms secondary to anxiety or depression [15]. ACT has also been used with EDs across the age spectrum, including adolescents [16] and adults with ED symptoms [17], and for the spectrum of ED issues (e.g., restriction, binge eating; Lillis et al., 2011). A few studies have also found ACT to aid with maintenance of treatment gains and avoidance of hospitalization [18,19]. 

Early evidence also supports increased acceptance (or more broadly, psychological flexibility) as a process of change in EDs. For example, studies have found low levels of acceptance are associated with ED symptom severity, and increased acceptance and body image flexibility (i.e., the ability to allow unwanted internal experiences, including negative thoughts/feelings about the body, without these experiences having undue influence over behavior) predicts symptom remission and moderates treatment response (e.g., BI-AAQ) [20,21]. There are also data suggesting that ED prevention programs that focus on body acceptance produce stronger effects relative to programs lacking this focus [1]. 

The vast majority of the evidence for ACT for ED studies have focused on individuals who were already exhibiting significant ED pathology. However, providing ACT skills early, when individuals have some ED signs and symptoms but before ED habits are formed, could alter illness trajectory. Yet, to our knowledge, there are not any ACT-based prevention or early-intervention programs that have been developed or tested for EDs. 

EDs characterized by body weight/shape concerns and maladaptive weight control (e.g., anorexia and bulimia nervosa spectrum) most commonly emerge between the ages of 16–20 years [1,22]. Previous studies have suggested that prevention and early intervention programs should thus target young women in this age range [1,22]. However, most young women with ED symptoms do not receive effective treatment [23]. There are several possible reasons for this, including that these individuals may choose low-threshold interventions instead of conventional health care provided for mental health problems [23]. Digital interventions have emerged in recent years and allow access to effective programs with lower costs, remove geographical constraints [24], and present the opportunity to cater to the interests and needs of youth with ED concerns. Such concerns include the desire for privacy or anonymity (due to high levels of guilt and shame) and access at all times of day and night [25]. Youth are also especially attracted to computer, console, and cell phone games, and there is a growing recognition that the use of digital gamification programs may be the preferred delivery method to reach today’s youths [26]. Gamification is the application of game design elements into processes and services in order to engage and motivate [27]. Although the influence of gamification strategies to attract users, increase motivation, and enhance learning is great, utilization of gamification components for behaviour change is still in its infancy. 

We developed an innovative digital, gamified ACT-based early-intervention program for adolescent girls at high risk for EDs (*AcceptME*; see [28] for a more extensive discussion of the treatment rationale). This study tests this program in a two-arm, randomized controlled trial. The *AcceptME* program presents with several innovations: (a) it was based on ACT theory and practices; (b) it used gamification principles to create a program appealing to youth and enhance learning through operant principles; (c) it used a storyline in which the user helps a digital character overcome their personal difficulties related to ED-related problems (e.g., body weight and shape concerns), providing opportunities for vicarious learning; and (d) it was a digitized early-intervention program for individuals exhibiting symptoms of ED who have not yet crossed the diagnostic threshold. There are few programs that are developed explicitly for digital media and early intervention [29]. We hypothesized that participants receiving the *AcceptME* program, relative to a waitlist control, would evidence (a) lower weight concerns; (b) fewer ED symptoms and behaviours; (c) improved quality of life; (d) decreased body dissatisfaction; and (e) greater psychological flexibility with body-image-related thoughts and feelings. 

## 2. Materials and Methods

### 2.1. Participants 

A total of 1050 young girls, 750 high-school students from twenty-five public high schools in Cyprus and 306 university students, aged 13 to 25 years (*M* = 16.8, *SD* = 2.8), participated in the study. The majority of the sample were Greek-Cypriot (91%), while 4% were Turkish Cypriot, and 0.5% were Maronite. Inclusion criteria were females aged 13–25 years, voluntary participation and parental consent for ages younger than 18 years, good working knowledge of the Greek language, and score of 52 or greater on the Weight Concern Scale at screening [30], indicating early signs of an ED and high risk for developing full-threshold ED. Consistent with the focus on ED early intervention, anyone who may have an active ED based on their score on the EDDS [31] was excluded and referred for an assessment of an eating disorder by a qualified professional. In this study, we chose to include only females as the chosen story-line was adapted for females, and given that eating disorders tend to appear somewhat differently in males, a different story-line would be more appropriate. While EDs can occur in males, women and girls (or female-identifying persons) continue to be at higher risk for these problems, with prevalence rates among women and girls far exceeding their male counterparts.

### 2.2. Procedure

This is a two-arm clinical trial that is registered at Clinicaltrial.gov (NCT03693911). Screening questionnaires were administered either manually or via online computerized administration (i.e., SurveyMonkey, www.surveymonkey.com/mp/audience, accessed on 25 January 2022) during class time and supervised by project researchers (N = 1050). Individuals identified as having a possible ED diagnosis (based on the EDDS) were referred to the Centre of Prevention and Treatment of Eating Disorders—Children and Adolescents Mental Health Services (only governmental clinic in the country offering these services). Individuals meeting criteria for participation (and their parents if <18 years of age; *n* = 292) were contacted via telephone and invited to participate in the trial. Of those invited, 92 agreed to participate and were randomized (2:1 randomization *AcceptME* vs. waitlist control). Of those, 62 were randomized to the *AcceptME* condition and were sent a text message with the webpage link, username, and password. Of those, 58 entered the program and completed the baseline assessment. Thirty were randomized into the control condition and were also asked to complete baseline measures. See Figure 1 for the CONSORT diagram of participant flow.

In addition to the baseline assessment, *AcceptME* participants completed assessments after each session, at end of intervention (EOI; i.e., immediately after completing Session 6), and at 1-month post-intervention follow-up. For every participant who completed the sixth session of *AcceptME*, one participant from the control group received a text message inviting her to complete the EOI assessment and providing the option to then enroll to receive the intervention.

#### *AcceptME* Program Description

*AcceptME* consisted of 6 sessions, and each had to be completed within the same day it was begun. Sessions were consecutive, and participants had to complete them in order (see [28]) for a video run-through of sessions. Three days after completion of a session, participants received a text message inviting them to continue to the next session. 

The *AcceptME* program portrayed the story of a young girl contemplating whether to enter a reality television fashion contest. Participants followed the main character through making the decision to enter the contest and faced situations eliciting common challenging situations, thoughts, and emotions, generally and especially related to her body image. A third-person perspective was used, where the participant viewed the events unfolding from the perspective of an observer. Participants were encouraged to assist the leading character progressing through the game by completing exercises teaching and applying ACT skills (e.g., acceptance, defusion, present moment awareness, values-consistent living) to progress in the contest and cope with distressing thoughts and feelings. Each session lasted approximately 30 minutes and included ACT experiential exercises and metaphors to teach core skills. The aim was for participants to develop new skills to be applied to their own life via helping the character achieve her goals. For a detailed description of the program content and development, see [28]. 

### 2.3. Measures 

Measures were completed by both the *AcceptME* and waitlist control group at baseline and EOI when the waitlist control group was offered access to the *AcceptME* program at that time. Participants in the *AcceptME* group completed the measures again at a 1-month post-intervention follow-up point. 

A demographic questionnaire assessed participants’ age, nationality/ethnicity, school, and year in school/university.

The Weight Concerns Scale (WCS) [30] consists of 5 questions assessing fear of weight gain, worry about weight and body shape, importance of weight, diet history, and perceived fatness. Scores > 52 are found to identify individuals at risk for later development (within the next 4 years) of an ED [30]. Killen and colleagues [30] reported satisfactory internal consistency (Cronbach’s alpha > 0.70), and in this sample, *a* = 0.78. 

The Eating Disorder Diagnostic Scale (EDDS) [31] is a 22-item self-report scale of ED symptoms. It has shown high agreement with diagnoses made with the Eating Disorder Examination (EDE) [32] and is used to identify individuals meeting ED diagnosis criteria with high validity and reliability [33,34]. In this study, it was used to identify individuals who possibly meet criteria for ED for exclusion purposes. Based on the scoring syntax of Stice et al. [31], scores indicating a possible ED diagnosis were excluded and referred for subsequent evaluation.

The Eating Disorder Examination Questionnaire (EDE-Q) [35] is a self-report version of the EDE interview, assessing frequency of key behavioural features and symptoms of eating disorders (i.e., number of times and days a behaviour has occurred in a 28-day period). Four subscales (Restraint, Weight Concern, Shape Concern, and Eating Concerns) and a global score are derived. Score range is from 0–6, and higher global and scale scores suggest greater severity of ED pathology and more symptoms. Numerous studies support its psychometric validity and reliability [33,34,35,36]. The Greek version of EDE-Q presents adequate internal consistency similar to previous studies, with Cronbach’s alphas for: Global = 0.78, Restraint = 0.74, Eating Concern = 0.91, Shape Concern = 0.91, and Weight Concern = 0.91 (Giovazolias, Tsaousis, and Vallianatou, 2012).

The Youth Quality of Life Instrument-Short Form (YQOL-SF) [37] is a self-report measure assessing the generic quality of life in youth aged 11–18 years. It includes 15 questions assessing domains of sense of self, social relationships, environment, and general quality of life. The response scale ranges from 0 (not at all) to 10 (completely). The scores are summed and then transformed to a 0 to 100 scale, with higher scores representing higher quality of life. It has good internal consistency (*α* = 0.80) for all four domains and for the total score [37]. In this study, the Greek YQOL-SF presented with good internal consistency *(a* = 0.75).

The Body Shape Questionnaire-8C (BSQ-8C) [38] is a self-reported, 8-question measure assessing body dissatisfaction and feelings of being fat. It presented with excellent treatment sensitivity to change in the course of therapy [39]. Questions refer to the participant’s body dissatisfaction over the past four weeks, rated on a 6-point scale from “never” to “always”. Higher scores indicate more body dissatisfaction. In the present study, the BSQ-8C Greek was found to have good internal consistency *(a* = 0.87), similar to previous reports (*α* = 0.93) [40].

The Body Image Acceptance and Action Questionnaire (BI-AAQ) [41] assesses psychological flexibility with body-image-related thoughts and feelings and specifically the ability to behave flexibly and effectively in the presence of thoughts and feelings about the body, without unnecessary attempts to avoid or escape these experiences. It is a unifactorial measure, and it demonstrated excellent internally consistent (*α* = 0.93), good concurrent validity, and good criterion-related validity [41]. The Greek version also demonstrated high internal consistency across two samples *(a* = 0.95) and was found to be a significant predictor of eating disorder behaviours after controlling for BMI and weight concern scores [42].

The Body Image Avoidance Questionnaire (BIAQ) [43] assesses behavioural tendencies of avoiding situations that trigger concerns about physical appearance. It also assesses other behaviours, such as frequent weighting and inspection in the mirror. It was found to have good internal consistency (Cronbach’s *α* = 0.89) and test-retest reliability *(r* = 0.87) [43]. Similarly, the Greek version in this study presented with good internal consistency *(a* = 0.87).

#### Statistical Analyses

Firstly, we assessed equivalence of the intervention and waitlist control groups at baseline on demographics and outcome variables using a two-tailed independent sample *t*-tests for age and multiple one-way, between-group analyses (independent variable was group: *AcceptME* vs. waitlist control) for dependent variables (i.e., treatment outcomes: weight concern, ED symptoms, body image flexibility, quality of life, body dissatisfaction, and body image avoidance). 

Individuals from the *AcceptME* group who completed all six sessions and the end-of-treatment assessment were considered completers, whereas participants who completed at least one *AcceptME* session but did not complete it in its entirety were considered non-completers. Multiple one-way, between-group analyses compared completers to non-completers for baseline outcome variables (i.e., weight concern, ED symptomatology, body image flexibility, quality of life, body dissatisfaction, and body image avoidance). There were no missing data, as the digital program required participants to complete each item before proceeding. 

To examine study hypotheses, a series of one-way Repeated Measures ANOVAs were conducted (IV: *AcceptME* vs. waitlist control; repeated factor was time: baseline vs. end-of-intervention). Intent-to-treat analyses (ITT), with the last observation carried forward (i.e., baseline) method on participants’ scores was also conducted for all significant findings. Finally, chi-square and risk analysis were conducted to examine the odds of falling into the at-risk group (WCS > 52) at end-of-intervention (EOI). Partial eta-square indices evaluated effects sizes as follows: ɳ^2^ < 0.09 (small effect), ɳ^2^ > 0.09 to <0.25 (medium effect), and ɳ^2^ > 0.25 (large effects). All statistical analyses were conducted using SPSS (IBM SPSS v.27).

## 3. Results

The mean age of participants (*n* = 89) enrolled in the trial was 15.30 years old (*SD* = 2.15; range: 13–22 years old). Sixty-three percent of those randomized completed post-assessment (*AcceptME* = 30, waitlist control = 28) and were included in the primary outcome analyses. Twenty-five of the participants (83% of the *AcceptME* group) competed the 1-month follow-up. 

### 3.1. Group Equivalence Prior to Intervention

There was no statistically significant difference in age between the two groups (*AcceptME*: *M* = 15.27, *SD* = 2.25; waitlist control: *M* = 15.09, *SD* = 1.89). Multiple one-way, between-group analyses indicated no significant differences (*p* > 0.050) between the *AcceptME* and waitlist control groups on baseline variables (i.e., weight concern, ED symptomatology, quality of life, body dissatisfaction, body image flexibility, and avoidance).

### 3.2. Comparison of Outcome and Process Variable at Baseline between Program Completers and Non-Completers

Multiple one-way, between-group analyses indicated no statistically significant differences (*p* > 0.050) between the completers and non-completers for baseline outcome variables (i.e., weight concern, ED symptomatology, quality of life, body dissatisfaction, and body image flexibility and avoidance). 

### 3.3. Repeated Measures Analysis of Variance of Group by Time on Outcomes

Means, standard deviations, and pre- and EOI-data of all outcomes for both groups are summarized in Table 1. A series of one-way Repeated Measures ANOVAs were conducted to examine the effects of time (2: baseline vs. end-of-intervention (EOI)) and group (2: *AcceptME* vs. waitlist control) on outcomes. 

There was a statistically significant interaction between time and group on participant’s WCS scores (*Cohen’s d* = 2.08—large effect). The main effect for time was not significant: *F*(1,54) = 2.97, *p* > 0.050, ɳ^2^ = 0.05. There was, however, a significant main effect for group: *F*(1,54) = 12.16, *p* < 0.001, ɳ^2^ = 0.95, with the *AcceptME* group presenting with lower weight concerns (M = 49.95, SD = 25.90) than the waitlist control group (*M* = 76.54, *SD* = 14.85). Single degree of freedom interaction contrasts showed that differences driving the interaction were those at EOI: *F*(1,54) = 24.40, *p* < 0.001, ɳ^2^ = 0.36 (see Figure 2). The ITT results were similar showing interaction effect, *F*(1,85) = 42.70, *p* < 0.010, ɳ^2^ = 0.33; main effect for time, *F*(1,85) = 4.21, *p* < 0.050, ɳ^2^ = 0.04; and main effect for group, *F*(1,85) = 15.76, *p* < 0.010, ɳ^2^ = 0.16. Single degree of freedom interaction contrasts showed significant differences between time, with the *AcceptME* group presenting significant decreases from baseline (*M* = 57.08; *SE* = 2.26) to EOI (*M* = 49.77; *SE* = 2.77) and the control group presenting significant increases (baseline: *M* = 62.00, *SE* = 3.12; EOI: *M* = 76.00, *SE* = 3.74). 

There was a statistically significant interaction between time and group on participants’ EDE-Q global scores (see Figure 3a). Single degree of freedom interaction contrasts showed that differences driving the interaction were those from baseline to EOI for the *AcceptME* condition only: *F*(1,55) = 9.99, *p* < 0.010, ɳ^2^ = 0.15 (waitlist control group changes from baseline to EOI, *p* > 0.050). Similarly, there was a statistically significant interaction between time and group on participants’ EDE shape concern subscale scores (see Figure 3b). Changes in the *AcceptME* condition from baseline to EOI and not in the waitlist control were those that drove the interaction effect based on single degree of freedom interaction contrasts, *F*(1,54) = 13.00, *p* < 0.010, ɳ^2^ = 0.19 (control group changes from baseline to EOI, *p* > 0.050). The interaction between time and group on participants’ EDE-Q weight concern subscale scores was marginally significant (see Figure 3c). Single degree of freedom contrasts showed that it was changes from baseline to EOI in the *AcceptME* group that drove this interaction effect, *F*(1,54) = 5.60, *p* = 0.020, ɳ^2^ = 0.10. No statistically significant interactions were found for EDE-Q restraint subscale scores and EDE-Q eating concern subscale scores. ITT analysis corroborated the findings where, for example, for the EDE-Q global score, there was a significant interaction, *F*(1,86) = 4.30, *p* < 0.050, ɳ^2^ = 0.05, and single degree of freedom interaction contrasts showing a significant decrease in ED symptoms in the *AcceptME* group from baseline (*M* = 2.20; *SE* = 0.11) to EOI (*M* = 1.91; *SE* = 0.11), whereas no differences were found for the control group (pre: *M* = 1.99, *SE* = 0.16; EOI: *M* = 2.08, *SE* = 0.15). 

No statistically significant interactions were found for body image flexibility. However, there was a significant time main effect *F*(1,54) = 6.24, *p* < 0.050, ɳ^2^ = 0.10 and single degree of freedom interaction contrasts showing that for the *AcceptME* group, there was a significant increase in body image flexibility at EOI compared to baseline (*p* < 0.010), and this was not noted for the waitlist control group (*p* > 0.050). ITT analysis corroborated these findings. 

There were no significant interactions for quality of life, body dissatisfaction, and body image avoidance. 

### 3.4. Analysis across Time (Baseline, EOI, 1-Month Follow-Up) for AcceptME Group Only

Means, standard deviations of baseline, EOI, and one-month follow-up assessments for outcomes are summarized in Table 2. A series of one-way Repeated Measures ANOVAs were conducted to examine the effects of time of the *AcceptME* group on outcomes.

The main effect of time was statistically significant on participants’ WCS scores. Pairwise comparisons indicated a significant difference on WCS mean scores between baseline and EOI (*p* < 0.010) as well as between baseline and 1-month follow-up (*p* < 0.001) but not between EOI and 1-month follow-up (*p* > 0.050). 

The main effect of time was statistically significant on participants’ EDE-Q global scores. Pairwise comparisons indicated a significant difference on the EDE-Q global scores between baseline and EOI (*p* < 0.050) and 1-month follow-up (*p* < 0.001), and EOI and 1-month follow-up (*p* < 0.010). Regarding subscales, there was a statistically significant time-effect for the EDE-Q eating concerns, *F*(2,23) = 12.14, *p* < 0.001, ɳ^2^ = 0.51. Pairwise comparisons indicated a significant difference between baseline and 1-month follow-up (*p* < 0.001) and between EOI and 1-month follow-up (*p* < 0.001) but not between baseline and EOI (*p* > 0.050). The time effect for the EDE-Q shape subscale scores was also significant, with the pairwise comparisons presenting differences between baseline and EOI (*p* < 0.050) and 1-month follow-up (*p* < 0.001), but there was no difference between EOI and 1-month follow-up (*p* > 0.050). Moreover, the main effect of time for EDE-Q weight concern subscale was statistically significant. Pairwise comparisons indicated significant differences between baseline and 1-month follow-up (*p* < 0.001) and EOI and 1-month follow-up (*p* < 0.050) but not between baseline and EOI (*p* > 0.05). The main effect for time was not statistically significant for the EDE-Q restraint scale. 

The repeated measures ANOVA for body dissatisfaction (BSQ-8C scores) across time for the *AcceptME* group was statistically significant. Pairwise comparisons indicated a significant difference between baseline and EOI scores (*p* < 0.010), baseline and 1-month follow-up (*p* < 0.001), and EOI and 1-month follow-up (*p* < 0.050). 

Similarly, the main effect of time was statistically significant on participant’s BI-AAQ scores. Pairwise comparisons indicated a significant difference between baseline and EOI scores (*p* < 0.050), baseline and 1-month follow-up (*p* < 0.001), and EOI and 1-month follow-up (*p* < 0.050). 

Statistical significance was also found across time on BIAQ scores. Pairwise comparisons indicated a significant decrease between the BIAQ scores between baseline and EOI (*p* < 0.001) and baseline and 1-month follow-up (*p* < 0.010). However, there was a significant increase between EOI and 1-month follow-up (*p* < 0.010). 

There was no significant time effect for quality of life.

### 3.5. Odds Ratios for Meeting Criteria for at-Risk at EOI

At EOI, 57.10% *(n* = 14) of participants and at -month follow-up, 72% (*n* = 18) of participants no longer met criteria for being at high risk for ED development (i.e., WCS score < 52; Killen et al., 1994). Chi-square analysis was statistically significant (*p* < 0.001), with a value of 16.75, and the risk analysis results showed the odds of falling into the at-risk group (WCS > 52) at EOI were 14.5 times higher for participants in the control vs. the *AcceptME* group (CI: 3.15, 66.67). 

## 4. Discussion

This is the first study to develop and test an innovative digital, gamified, ACT-based ED early-intervention program for youth showing signs and symptoms of an ED and being at high risk for the development of full syndrome. Consistent with our hypothesis, young women who participated in the *AcceptME* program had significantly lower weight and shape concerns at the end of the program in comparison to participants in the waitlist control group with large effect sizes. This is consistent with at least one other acceptance and mindfulness-based digital program for EDs. Atkinson and colleagues [44], for example, assessed a mindfulness-based intervention to reduce the risk for ED development, and their results showed significant reductions in WCS with large effect size. These effect sizes exceed the medium effect sizes reported by other programs, such as the digital cognitive dissonance program [45,46]. Stice et al. [47,48] reported larger reductions in ED risk in their Body Project with effect sizes ranging between small to medium effect sizes (*d* = 0.16–0.59), including their online version of this intervention (eBody Project). In the current study, effects on weight concerns were larger (*d* = 2.08). Other early-intervention studies are based on traditional CBT, with very few digital options. One exception is Student Bodies [47], an online CBT intervention that has been used for selective prevention and indicated prevention/early intervention. It includes a whole host of targets and strategies (e.g., self-monitoring, cognitive challenging, psychoeducation, emotion regulation, meal planning, etc.). Student Bodies leads to mild to moderate improvement in ED psychopathology (small effect sizes). However, effects are primarily relative to waitlist controls and for some but not all ED behaviours (e.g., bingeing vs. restricting) or were not maintained at follow-up [48,49] or have been nil in intent-to-treat analyses [50]. Other early-intervention approaches include programs, such as Es(s)pirit [51] and ProYouth [52]; however, these programs are mostly focused on education and support (with some tailored “feedback”) and referral for individuals with more substantial symptoms. The present study utilized a gamified, tailored ACT approach for early-intervention, and findings are overall promising and support the continued development of such programs to reduce ED symptoms in young females at risk for EDs.

Most young women who completed the *AcceptME* intervention scored below the cut-off score for being in the at-risk category for developing an ED in the next four years at end-of-intervention (57.10%; WCS score < 52) [30]. Interestingly, 72% of intervention completers scored below this cut-off at the 1-month follow-up. The average mean score on the WCS (*M* = 45.95) was overall below the cut-off with evidenced additional reductions in WCS scores (*M =* 40.80) at the 1-month follow-up. In contrast, young women in the waitlist control group had a statistically significant increase in weight and shape concerns (*M =* 67.14), with only 7.1% scoring below the at-risk cut-off at EOI, revealing 14.5 times higher odds of being at-risk of developing an ED compared to the *AcceptME* group.

The *AcceptME* group also had significantly lower scores on the global EDE-Q as well as the weight and shape concern subscales. These subscales assess some of the cognitive features of ED symptoms, suggesting that participants became less preoccupied with weight and shape after their participation in the program. Improvements in ED symptoms appeared to be maintained at the 1-month follow-up period with large effect sizes. However, these findings should be interpreted with caution given the lack of comparison group for the 1-month follow-up. Taken together, the results of the current study support that ACT-based skills learned in this novel digital, gamified ED early-intervention program can reduce ED symptomatology and parameters associated with the risk for ED development. 

For some outcomes, there were no interaction effects but significant time effects for the *AcceptME* group, with continued improvement over time to the 1-month follow-up. Body image flexibility notably showed continued improvement over time. This may reflect additional consolidation of acceptance-based concepts, consistent with other studies that have shown delayed but potentially more persistent effects in ACT interventions. However, long-term follow-up with a control condition is needed. 

No significant effects were noted for quality of life. The scores reported on the YQOL-SF were close to those reported for general population of adolescents who do not present significant psychopathology (e.g., [53,54]); thus, it may be that participants in the present study did not experience significant quality of life impacts, and thus, there was not much room for improvement. The impacts on quality of life of youth presenting at high risk for developing an ED warrants further research examination. 

This is the first study to assess this kind of a digital, gamified ACT early-intervention program for individuals at high risk for developing an ED and also utilizing a non-threatening vicarious learning approach. The large effect sizes for key outcomes is of note. Unlike some prevention and early-intervention programs, this program, based on ACT principles, takes into consideration and attempts to alter the function that ED symptoms serve for the individual. This includes the negative functions, such as removal of unwanted internal experiences, as well as the positive experiences of mastery or pride with successful eating and weight control [55,56]. The program helps the individual recognize these different functions how efforts to control or manipulate their internal experiences may work in the short term but be ineffective in the long-run or for the individuals’ personal values. Through the story-line and exercises, individuals learn new ways of responding to unwanted thoughts and feelings, and via engagement with values, contact alternative reinforcers. There are different approaches to addressing irrational beliefs about eating and body weight/shape central to EDs. One is to try to change at the content level (make more rational, etc., e.g., [57]), while another is to increase cognitive and behavioural flexibility (or how one relates and responds to their thoughts/feelings, e.g., [56]). The current intervention takes that latter approach. This approach may be well matched to EDs given that individuals with EDs (particularly anorexia) tend to be more rigid, and beliefs tend to be resistant to change. Targeting inflexibility might also address some of the neurodevelopmental differences (see [58]) that may be implicated in ED development or contribute to poorer outcomes (particularly in AN). 

An additional strength of the intervention is the digital format, which allowed individuals to participate at their own time and place. Furthermore, since the program is web-based delivered, it could be easily implemented and disseminated to potential users in the future. Once refined, the digital prevention program can be considered by high schools, colleges, or universities to reduce ED symptom progression. Further research and replication and implementation research in other contexts is important and warranted. 

Although these results are promising, future investigation is needed to clarify the specific treatment components (e.g., increased motivation, values clarification, psychological flexibility, cognitive defusion) responsible for therapeutic effects with explicit exploration of the proposed mechanisms of change.

### Limitations of the Study

A key limitation of the study was that group comparisons were only able to be conducted at end-of-intervention, as participants in the control group were provided access to the treatment at that point. It would be beneficial to evaluate the *AcceptME* program in a study with a longer-term follow-up. Longer-term follow-up would have been especially useful, as the results suggested additional improvements in psychological flexibility and ED symptoms to occur between the post and the one-month follow-up time points. The study was also limited by the use of waitlist control group instead of another active intervention for comparison. The use of another treatment as a control was difficult in the present study, as no similar already established digital intervention in Greek language was found. Consistent with the early-intervention focus of the study, participants were categorized as being at high risk based on responses to a standardized test and were not screened or assessed clinically by a professional. Future studies can expand current work to include a clinical assessment. Finally, the sample was relatively homogenous with regards to ethnicity, suggesting that care should be taken in generalizing the results to more diverse populations. It would be beneficial to have the program translated in other languages so that experts on ACT and ED can review and provide fidelity evidence for the program and to further assess the program in other ethnic, racial, and cultural groups. Another limitation involves the number of individuals who were recognized to be at high risk but did not consent to take part in the intervention trial. Although it was expected that this is a difficult population to engage in any sort of intervention program, future attempts need to be made to motivate individuals deemed to be at risk to engage in such intervention programs. In addition, non-completion once individuals enter an intervention is another important issue that needs to be considered and a limitation of the study. Though the percent of non-completion is similar to previous similar studies [59,60,61], and ITT analyses did corroborate the findings of the trial, caution in interpretation is needed. Despite these limitations, the results of the study suggest that this is a promising new approach even if further research is necessary.

## 5. Conclusions

In conclusion, the current study raises the possibility that an ACT-based digital gamified early-intervention program may be effective for young women with early signs and symptoms of an ED and at risk for ED progression. More research is needed to replicate and extend these early findings.

## Figures and Tables

**Figure 1 jcm-11-01775-f001:**
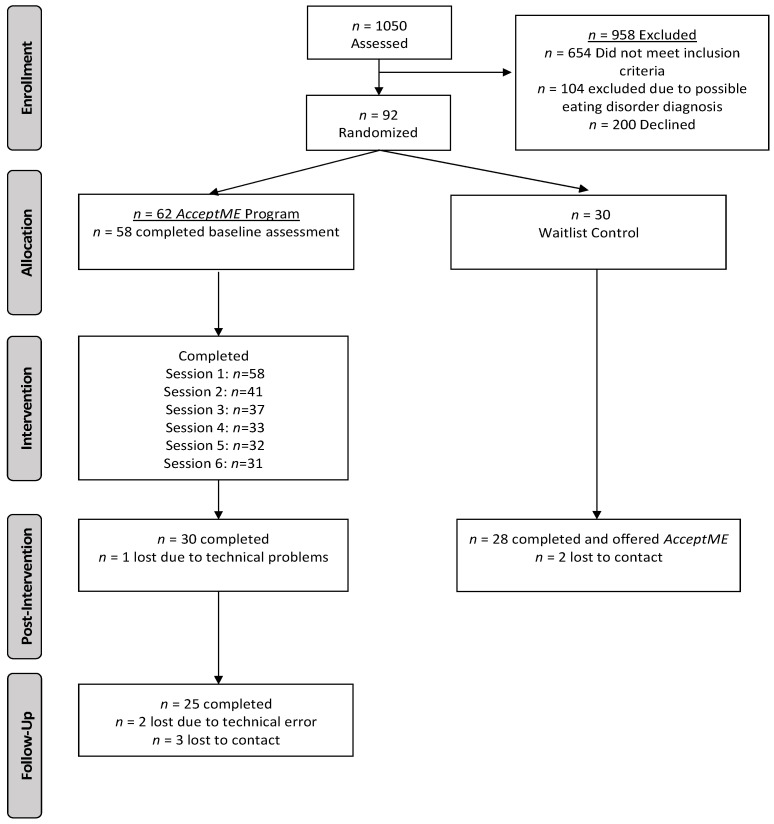
CONSORT flow diagram.

**Figure 2 jcm-11-01775-f002:**
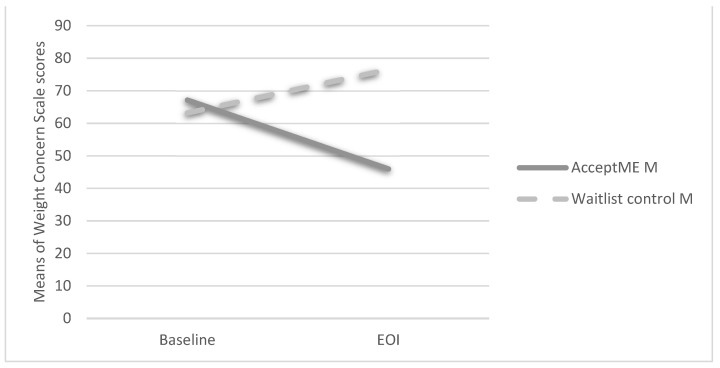
Group by Time Interaction of Weight Concern Scale scores. Note: EOI = End-of-intervention time point of assessment; M = Mean value.

**Figure 3 jcm-11-01775-f003:**
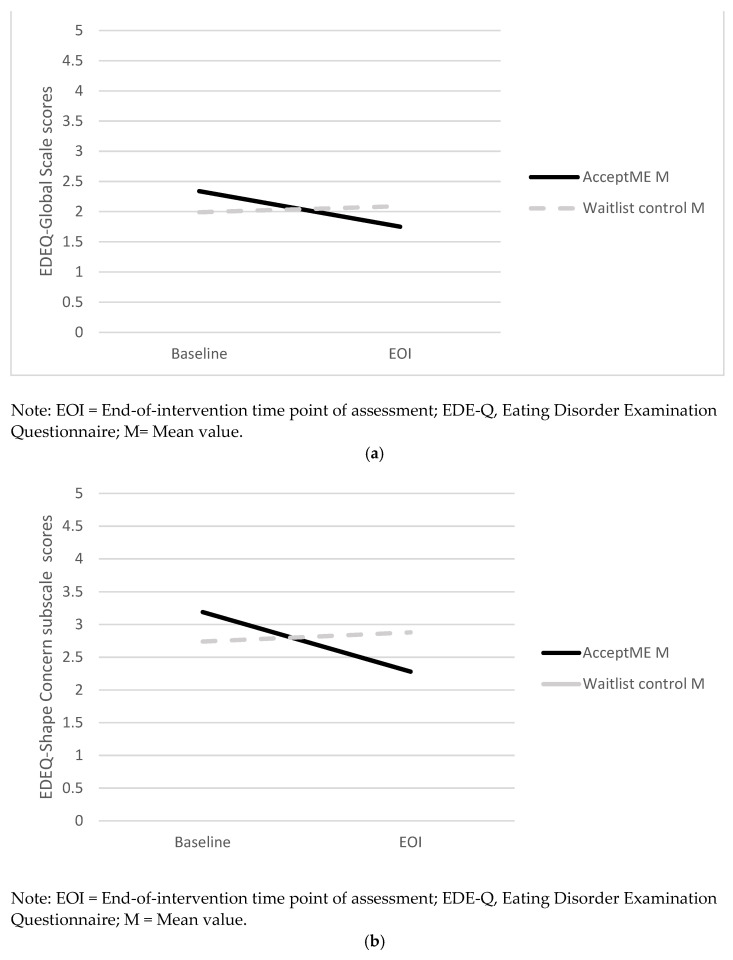
(**a**) Group by Time Interaction of EDEQ—Global score. (**b**) Group by Time Interaction of EDEQ—Shape concern subscale scores. (**c**) Group by Time Interaction of EDEQ—Weight concern subscale scores.

**Table 1 jcm-11-01775-t001:** Means and standard deviations of outcome at baseline and end-of-intervention by group (*AcceptME* vs. waitlist control).

		*AcceptME* Group		Control Group		Interaction Effects
		Baseline M (SD) *n* = 59	EOI M (SD) *n* = 29	Baseline M (SD) *n* = 29	EOI M (SD) *n* = 28	
WCS		67.14 (11.93)	45.95 (25.90)	63.19 (8.58)	76.54 (14.85)	***F*_(1,54)_ = 56.67, *p* < 0.001, ɳ^2^ = 0.52**
% score < 52 high-risk cut-off		0%	57.10%	0%	7.10%	
EDE-Q						
	Restraint Subscale	1.82 (1.34)	1.37 (1.06)	1.64 (1.19)	1.70 (0.95)	*F*_(1,54)_ = 3.81, *p* = 0.058, ɳ^2^ = 0.07
	Eating Concern Subscale	1.59 (1.06)	1.22 (1.02)	1.37 (0.92)	1.27 (0.84)	*F*_(1,54)_*=* 1.08, *p* > 0.050, ɳ^2^ *=* 0.02
	Shape Concern Subscale	3.18 (1.21)	2.28 (1.33)	2.74 (1.07)	2.88 (1.05)	***F*_(1,54)_ = 8.49, *p* < 0.010, ɳ^2^ = 0.14**
	Weight Concern Subscale	2.76 (1.07)	2.17 (1.01)	2.45 (1.00)	2.56 (1.01)	***F*_(1,54)_ = 3.81, *p* = 0.052, ɳ^2^ = 0.07**
	Global	2.34 (0.93)	1.75 (0.99)	1.99 (0.91)	2.09 (0.66)	***F*_(1,55)_ = 6.80, *p* = 0.010, ɳ^2^ = 0.11**
YQOL-SF		66.06 (10.05)	62.71 (22.57)	68.06 (11.86)	68.21 (10.10)	*F*_(1,49)_ = 0.46, *p* > 0.050, ɳ^2^ = 0.01
BSQ-8C		29.93 (8.35)	23.34 (11.27)	26.54 (10.01)	26.27 (11.35)	*F*_(1,49)_ = 3.78, *p* = 0.058, ɳ^2^ = 0.07
BI-AAQ		48.45 (16.90)	56.76 (15.54)	48.41 (13.32)	50.85 (15.10)	*F*_(1,54)_ = 1.86, *p* > 0.050, ɳ^2^ = 0.03
BIAQ		35.44 (11.62)	19.75 (8.58)	37.54 (18.45)	23.86 (11.05)	*F*_(1,49)_ = 0.65, *p* > 0.050, ɳ^2^ = 0.01

Note: EOI, end-of-intervention time point of assessment; WCS, Weight Concerns Scale; EDE-Q, Eating Disorder Examination Questionnaire; YQOL-SF, Youth Quality of Life—Short Form; BSQ-8C, Body Shape Questionnaire—8C; BI-AAQ, Body Image-Acceptance and Action Questionnaire; BIAQ, Body Image Avoidance Questionnaire; Bolded statistics present significant findings.

**Table 2 jcm-11-01775-t002:** Means and standard deviations of outcomes at baseline, EOI, and one-month follow-up for the *AcceptME* group.

		Baseline M (SD) *n* = 25	EOI M (SD) *n* = 25	One-Month Follow-Up M (SD) *n* = 25	F-Test
WCS		56.27 (20.24) ^a^	44.33 (25.84) ^b^	40.80 (21.81) ^b^	** *F* ** ** _(2,23)_ ** **= 9.30, *p* < 0.001,** **ɳ** ** ^2^ ** **= 0.45**
EDE-Q					
	Restraint Subscale	1.57 (1.18)	1.39 (1.05)	1.10 (1.01)	*F*_(2,23)_ = 2.09, *p* > 0.050, ɳ^2^ = 0.15
	Eating Concern Subscale	1.49 (1.02) ^a^	1.25 (1.06) ^a^	0.75 (0.87) ^b^	** *F* ** ** _(2,23)_ ** **= 12.14, *p* < 0.001,** **ɳ** ** ^2^ ** **= 0.51**
	Shape Concern Subscale	3.11 (1.16) ^a^	2.42 (1.34) ^b^	2.11 (1.12) ^b^	** *F* ** ** _(2,23)_ ** **= 9.65, *p* < 0.001,** **ɳ** ** ^2^ ** **= 0.46**
	Weight Concern Subscale	2.58 (0.99)	2.17 (1.01)	1.87 (0.76)	** *F* ** ** _(2,23)_ ** **= 10.20, *p* < 0.001,** **ɳ** ** ^2^ ** **= 0.47**
	Global	2.20 (0.84) ^a^	1.76 (0.99) ^b^	1.46 (0.81) ^bc^	** *F* ** ** _(2,23)_ ** **= 14.93, *p* < 0.001,** **ɳ** ** ^2^ ** **= 0.57**
YQOL-SF		66.07 (10.05)	62.71 (22.58)	68.06 (11.86)	*F*_(2,23)_ = 7.48, *p* > 0.050, ɳ^2^ = 0.03
BSQ-8C		30.12 (7.92) ^a^	24.12 (11.56) ^b^	21.56 (8.91) ^bc^	** *F* ** ** _(2,23)_ ** **= 13.38, *p* < 0.001,** **ɳ** ** ^2^ ** **= 0.54**
BI-AAQ		49.36 (17.01) ^a^	57.24 (16.20) ^b^	63.44 (14.34) ^bc^	** *F* ** ** _(2,23)_ ** **= 10.40, *p* < 0.001,** **ɳ** ** ^2^ ** **= 0.48**
BIAQ		35.44 (11.32) ^a^	20.68 (8.53) ^b^	26.48 (11.12) ^bc^	** *F* ** ** _(2,23)_ ** **= 26.64, *p* < 0.001,** **ɳ** ** ^2^ ** **= 0.70**

Note 1: ^a^ significantly different from ^b^ and ^b^ significantly different from ^c^. Note 2: EOI, end-of-intervention; WCS, Weight Concerns Scale; EDE-Q, Eating Disorder Examination Questionnaire; YQOL-SF, Youth Quality of Life Instrument—Short Form; BSQ-8C, Body Shape Questionnaire-8C; BI-AAQ, Body Image-Acceptance and Action Questionnaire; BIAQ, Body Image Avoidance Questionnaire. Bolded statistics present significant findings.

## Data Availability

Data are available from the corresponding author upon request.

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
