# Peer review of "Randomized Clinical Trial Evaluating AcceptME—A Digital Gamified Acceptance and Commitment Early Intervention Program for Individuals at High Risk for Eating Disorders"

_jcm, 2022, doi:10.3390/jcm11071775_

Round 1

Reviewer 1 Report

The manuscript presents some interesting results from a randomized-controlled trial conducted with the aim of evaluating the effects of an innovative gamified ACT-based intervention program for the prevention of eating disorders. While the manuscript is correct in writing and appropriate in methodology, some minor revisions might help to improve its quality and overall clarity, especially in the Methods and Results sections.

  • The introduction is well written and it accurately presents the literature about the application of ACT in eating disorders, as well as the possible utility of applying digital intervention in this field. The introduction is also appropriate in length and no further revision is needed. However, at line 107, page 3, I would suggest to substitute the term “body image flexibility” with “psychological flexibility with body-image related thoughts and feelings” to avoid confusion and make it the same as the term used when describing the questionnaire to assess this variable (BI-AAQ).
  • The paragraph Statistical Analyses may benefit from a revision, as it can be a bit confusing for the readers. In particular, I suggest the authors to better specify the dependent and independent variables used for each analysis. For example, instead of writing that “To examine study hypotheses, a series of one-way Repeated Measures ANOVAs were conducted” (lines227-228, page 5), it would be better to specify which specific groups, times and variables were included in the ANOVAs. I would also advise the authors to replace the term “treatment outcomes” in this paragraph with an explicit list of variables that were considered in the analyses.
  • Authors should also make sure that the statistical analyses reported in this paragraph follow the same order as the results presented in the paragraphs they used for dividing the Results section. For example, while the authors stated (in the Statistical Analyses paragraph) that the participants were divided into completers and non-completers, they did not mention the fact that statistical analyses were also performed to compare the two groups. The results of these analyses were instead presented in the paragraph 3.2 of the Results section.
  • In Table 1 and 2 authors should report the interaction effects and the results of the F-tests also for non-significant comparisons. If the authors wish to better distinguish between significant and non-significant results in the tables, this can be done in other ways (such as with bold text or asterisks). 

Author Response

Reviewer 1

  • Line 107 p3: I would suggest to substitute the term “body image flexibility” with “psychological flexibility with body-image related thoughts and feelings” to avoid confusion and make it the same as the term used when describing the questionnaire to assess this variable(BI-AAQ).

The term has been changed as suggested.

  • The paragraph Statistical Analyses may benefit from a revision, as it can be a bit confusing for the In particular, I suggest the authors to better specify the dependent and independent variables used for each analysis. For example, instead of writing that “To examine study hypotheses, a series ofone-way Repeated Measures ANOVAs were conducted” (lines227-228, page 5), it would be better to specify which specific groups, times and variables were included in the ANOVAs. I would also advise the authors to replace the term “treatment outcomes” in this paragraph with an explicit list of variables that were considered in the analyses.

The paragraph on Statistical Analyses was changed as suggested. We added the independent and dependent variables, we specified the treatment outcome variables, and explicated what the between and within group variables were for the repeated measures ANOVA.

  • Authors should also make sure that the statistical analyses reported in this paragraph follow the same order as the results presented in the paragraphs they used for dividing the Results section. For example, while the authors stated (in the Statistical Analyses paragraph) that the participants were divided into completers and non-completers, they did not mention the fact that statistical analyses were also performed to compare the two groups. The results of these analyses were instead presented in the paragraph 3.2 of the Results section.

This was an oversight and we have ensured to mirror in the statistical analyses what was done in the results section.

  • In Table 1 and 2 authors should report the interaction effects and the results of the F-tests also for non-significant comparisons. If the authors wish to better distinguish between significant and non-significant results in the tables, this can be done in other ways (such as with bold text or asterisks)

As suggested, we included the non-significant comparisons and highlighted significant findings by bolding them on Tables 1 & 2.

Reviewer 2 Report

I have carefully revised the article “Randomized clinical trial evaluating a digital gamified 2 Acceptance and Commitment early intervention program for 3 individuals at high-risk for eating disorders”.

Major points to consider

A major limitation of this study is that, apparently, standardized tests were administered to subjects who have never been clinically screened by a physician or a psychologist. Since the article has been submitted for consideration to a Journal of Clinical Medicine, this should be consistently acknowledged as a limitation across the manuscript, especially in passages like line 117 (“Exclusion criteria were: possible current ED diagnosis as assessed 117 by the Eating Disorder Diagnostic Scale (EDDS)”). It is correct that the authors use the word “possible”, but the diagnosis of ED is primarily clinical and independent from any single questionnaire, thus these distinctions should be emphasized.

Line 76: “EDs most commonly emerge between the ages of 16-20 years”. This sentence is somewhat general and not totally correct. The DSM-5 recognizes different types of ED, including Avoidance/Restrictive Food Intake Disorder. Since the risk/possible onset of ED represents a theoretical prerequisite of this study, the authors should dedicate a few lines here to disentangle possible differences across diverse ED on the age at onset and the response to cognitive-behavioral interventions.

The authors should invest a small paragraph to clearly indicate why they deliberately excluded males (or any non-girl subject) from this study.

Paragraphs 3.3 and 3.4 in the results section, despite a clear illustration of the original findings, may result in excessive to the reader. I suggest that these results should be moved to 2 tables. Given the high number of figures used in this manuscript, the use of supplementary materials or merging different figures should be considered.

The discussion section is a bit general and should be more focused. The authors should also emphasize the potential clinical factors influencing the intervention here reported. To cite some non-exclusive examples published in this journal, recent relevant roles have been documented for irrational beliefs (Tecuta L, Gardini V, Schumann R, Ballardini D, Tomba E. Irrational Beliefs and Their Role in Specific and Non-Specific Eating Disorder Symptomatology and Cognitive Reappraisal in Eating Disorders. J Clin Med. 2021 Aug 11;10(16):3525. doi: 10.3390/jcm10163525. PMID: 34441821; PMCID: PMC8397039.), neurodevelopmental disorders and traits (Pruccoli J, Rosa S, Cesaroni CA, Malaspina E, Parmeggiani A. Association among Autistic Traits, Treatment Intensity and Outcomes in Adolescents with Anorexia Nervosa: Preliminary Results. J Clin Med. 2021 Aug 16;10(16):3605. doi: 10.3390/jcm10163605. PMID: 34441899; PMCID: PMC8397224.) and expressed emotions (Philipp J, Truttmann S, Zeiler M, Franta C, Wittek T, Schöfbeck G, Mitterer M, Mairhofer D, Zanko A, Imgart H, Auer-Welsbach E, Treasure J, Wagner G, Karwautz AFK. Reduction of High Expressed Emotion and Treatment Outcomes in Anorexia Nervosa-Caregivers' and Adolescents' Perspective. J Clin Med. 2020 Jun 27;9(7):2021. doi: 10.3390/jcm9072021. PMID: 32605074; PMCID: PMC7409203.). These or other contributions could provide a more “clinical” background to an already interesting study.

Minor points to consider

Abbreviations should be spelled when they are reported for the first time in the manuscript, independently from the abstract (e.g., ED, CBT).

p values are sometimes reported this way “p<.01”, sometimes this way “p=.01” and sometimes this way “p<.001”. The last of the three (3 decimals after the dot) is preferred and should be maintained across the manuscript.

Author Response

Reviewer 2

I have carefully revised the article “Randomized clinical trial evaluating a digital gamified Acceptance and Commitment early intervention program for individuals at high-risk for eating disorders”.

Major points to consider

  • A major limitation of this study is that, apparently, standardized tests were administered to subjects who have never been clinically screened by a physician or a psychologist. Since the article has been submitted for consideration to a Journal of Clinical Medicine, this should be consistently acknowledged as a limitation across the manuscript, especially in passages like line117 (“Exclusion criteria were: possible current ED diagnosis as assessed by the Eating Disorder Diagnostic Scale (EDDS)”).It is correct that the authors use the word “possible”, but the diagnosis of ED is primarily clinical and independent from any single questionnaire, thus these distinctions should be emphasized.

We acknowledged as suggested the lack of a clinical assessment as a limitation to the study. We have changed the wording in line 117 to be more clear about what was done regarding the exclusionary criteria.

  • Line 76: “EDs most commonly emerge between the ages of 16-20 years”. This sentence is somewhat general and not totally correct. The DSM-5 recognizes different types of ED, including Avoidance/Restrictive Food Intake Disorder. Since therisk/possible onset of ED represents a theoretical prerequisite of this study, the authors should dedicate a few lines here to disentangle possible differences across diverse ED on the age at onset and the response to cognitive-behavioral interventions.

Changes were made to this sentence based on the reviewers’ comment and to explain this statement better.

We added the following:

EDs characterized by body weight/shape concerns and maladaptive weight control (e.g., anorexia and bulimia nervosa spectrum) most commonly emerge between the ages of 16-20 years [1,22].

  • The authors should invest a small paragraph to clearly indicate why they deliberately excluded males (or any non-girl subject) from this study.

As suggested, we added information regarding this choice to focus this study only on females. The following was added:

In this study we chose to include only females as the chosen story-line was adapted for females and given that eating disorders tend to appear somewhat differently in males, a different story-line would be more appropriate. While EDs can occur in males, women and girls (or female-identifying persons) continue to be at higher risk for these problems, with prevalence rates among women and girls far exceeding their male counterparts.

  • Paragraphs 3.3 and 3.4 in the results section, despite a clear illustration of the original findings, may result in excessive to the reader. I suggest that these results should be moved to 2 tables. Given the high number of figures used in this manuscript, the use of supplementary materials or merging different figures should be considered.

We added the Fs etc for significant findings as suggested by Reviewer 1, deleted the Fs presented in the tables from also appearing in the text, but retained other information in the text (rather than adding it to the tables) to prevent the tables from becoming too large and difficult to review. We decreased the number of figures by deleting the ones presenting the AcceptME group declining over time which is easily conveyed by the provided means at baseline, EOI and 1 month follow-up.

  • The discussion section is a bit general and should be more focused. The authors should also emphasize the potential clinical factors influencing the intervention here reported. To cite some non-exclusive examples published in this journal, recent relevant roles have been documented for irrational beliefs (Tecuta L, Gardini V, Schumann R, Ballardini D, Tomba E.Irrational Beliefs and Their Role in Specific and Non-SpecificEating Disorder Symptomatology and Cognitive Reappraisal inEating Disorders. J Clin Med. 2021 Aug 11;10(16):3525. doi:10.3390/jcm10163525. PMID: 34441821; PMCID:PMC8397039.), neurodevelopmental disorders and traits (Pruccoli J, Rosa S, Cesaroni CA, Malaspina E, Parmeggiani A.Association among Autistic Traits, Treatment Intensity and Outcomes in Adolescents with Anorexia Nervosa: PreliminaryResults. J Clin Med. 2021 Aug 16;10(16):3605. doi:10.3390/jcm10163605. PMID: 34441899; PMCID:PMC8397224.) and expressed emotions (Philipp J, Truttmann S,Zeiler M, Franta C, Wittek T, Schöfbeck G, Mitterer M, MairhoferD, Zanko A, Imgart H, Auer-Welsbach E, Treasure J, Wagner G, Karwautz AFK. Reduction of High Expressed Emotion andTreatment Outcomes in Anorexia Nervosa-Caregivers' andAdolescents' Perspective. J Clin Med. 2020 Jun 27;9(7):2021.doi: 10.3390/jcm9072021. PMID: 32605074; PMCID:PMC7409203.). These or other contributions could provide a more “clinical” background to an already interesting study.

We have made additions to the discussion to make it more focused and “clinical” as suggested and included some of the references proposed.

Minor points to consider

  • Abbreviations should be spelled when they are reported for the first time in the manuscript, independently from the abstract (e.g., ED, CBT).

We have ensured to spell out all the abbreviations the first time they appear in the manuscript independent of the abstract.

  • p values are sometimes reported this way “p<.01”, sometimesthis way “p=.01” and sometimes this way “p<.001”. The last of the three (3 decimals after the dot) is preferred and should bemaintained across the manuscript.

As suggested, we have adopted 3 decimal places after the dot throughout when reporting p values.

Reviewer 3 Report

The title of the article is consistent with its content.

There is no clear division into sections in the abstract: introduction, material and method, results, conclusions.

Introduction: covers basic concepts and problems related to Eating Disorders (ED): definition, causes, symptoms and treatment options. The authors point out that influencing the ability to direct thoughts, accepting difficult thoughts and emotions, and engaging in effective health activities is an important element of therapy (cognitive behavioral theory of ED - CBT). However, more effective is Acceptance and Commitment Therapy (ACT) aimed at learning to react to your own thoughts and feelings. ACT helps in maintaining the benefits of treatment and avoiding hospitalization. Currently, it is possible to conduct therapy based on technical and digital achievements. Conducting digital therapy satisfies the desire for privacy, anonymity and is available at any time of the day or night. Young people generally like computer games, console games and games on mobile phones. The authors developed a digital therapeutic program (AcceptME) based on the assumptions of ACT therapy for adolescent girls at high risk of eating disorders. The authors hypothesized that the participants of the AcceptME digital therapy program, compared to the control group, would show less severe symptoms of eating disorders, improved quality of life, and reduced dissatisfaction with their own bodies.

Material and methods: a total of 1050 women aged 13-25 were enrolled in the study, with a score of at least 52, obtained on the basis of the Weight Concern Scale (a result indicating the presence of early symptoms of eating disorders and a high risk of developing complete disorders of the nutrition). The study was registered at Clinicaltrial.gov. People suspected of having eating disorders were referred for treatment in a specialized unit. In total, the full set of tests and therapy with AcceptME was completed by 58 people qualified for the tests (main group) and a full set of tests of 30 people (control group). The methodology of therapy using AcceptMe (6 therapeutic sessions) was characterized, a set of questionnaires used for the study included: a demographic questionnaire, the Weight Concerns Scale (WCS), The Eating Disorder Diagnostic Scale (EDDS), The Eating Disorder Examination Questionnaire (EDE-Q), The Youth Quality of Life Instrument-Short Form (YQOL-SF). The Body Image Acceptance and Action Questionnaire (BI-AAQ), The Body Image Avoidance Questionnaire (BIAQ) (measurements were performed before the first treatment session and after each subsequent treatment session).

The results are comprehensively discussed, presented in figures and tables. However, there are no abbreviations under the figures. For example, most participants of treatment sessions using the AcceptMe program achieved a WCS score of <52 at the end of the intervention (57.1%) compared to the control group (7.1%).

Discussion: seems too laconic. Eight out of 49 references were used to compose the discussion. I propose to broaden the discussion by comparing the results presented in five different articles published in the last five years, at least. The method used in the treatment of eating disorders should be taken into account and the significance of the change in the parameters studied should be presented. The effectiveness of using digital therapy in the treatment of other disorders can be demonstrated.

The described limitations of the research are satisfactory.

Author Response

Reviewer 3

Introduction: covers basic concepts and problems related to Eating Disorders (ED): definition, causes, symptoms and treatment options. The authors point out that influencing the ability to direct thoughts, accepting difficult thoughts and emotions, and engaging in effective health activities is an important element of therapy (cognitive behavioral theory of ED -CBT). However, more effective is Acceptance and Commitment Therapy (ACT) aimed at learning to react to your own thoughts and feelings. ACT helps in maintaining the benefits of treatment and avoiding hospitalization. Currently, it is possible to conduct therapy based on technical and digital achievements. Conducting digital therapy satisfies the desire for privacy, anonymity and is available at any time of the day or night. Young people generally like computer games, console games and games on mobile phones. The authors developed a digital therapeutic program (AcceptME) based on the assumptions of ACT therapy for adolescent girls at high risk of eating disorders. The authors hypothesized that the participants of the AcceptME digital therapy program, compared to the control group, would show less severe symptoms of eating disorders, improved quality of life, and reduced dissatisfaction with their own bodies.

Material and methods: a total of 1050 women aged 13-25 were enrolled in the study, with a score of at least 52, obtained on the basis of the Weight Concern Scale (a result indicating the presence of early symptoms of eating disorders and a high risk of developing complete disorders of the nutrition). The study was registered at Clinicaltrial.gov. People suspected of having eating disorders were referred for treatment in a specialized unit. In total, the full set of tests and therapy with AcceptME was completed by 58 people qualified for the tests (main group) and a full set of tests of 30 people (control group). The methodology of therapy using AcceptMe (6 therapeutic sessions) was characterized, a set of questionnaires used for the study included: a demographic questionnaire, the Weight Concerns Scale (WCS), The Eating Disorder Diagnostic Scale (EDDS),The Eating Disorder Examination Questionnaire (EDE-Q), The Youth Quality of Life Instrument-Short Form (YQOL-SF). The Body Image Acceptance and Action Questionnaire (BI-AAQ),The Body Image Avoidance Questionnaire (BIAQ)(measurements were performed before the first treatment session and after each subsequent treatment session).

The results are comprehensively discussed, presented in figures and tables. However, there are no abbreviations under the figures. For example, most participants of treatment sessions using the AcceptMe program achieved a WCS score of <52 at the end of the intervention (57.1%) compared to the control group (7.1%).

As suggested, a note presenting the abbreviations was added under each of the figures.

Discussion: seems too laconic. Eight out of 49 references were used to compose the discussion. I propose to broaden the discussion by comparing the results presented in five different articles published in the last five years, at least. The method used in the treatment of eating disorders should be taken into account and the significance of the change in the parameters studied should be presented. The effectiveness of using digital therapy in the treatment of other disorders can be demonstrated.

The described limitations of the research are satisfactory.

We have made changes to the discussion to broaden it and compare the results to other studies, and especially recent published articles.